# Phase Instability, Oxygen Desorption and Related Properties in Cu-Based Perovskites Modified by Highly Charged Cations

Roman A. Shishkin * , Alexey Yu. Suntsov  and Mikhael O. Kalinkin

Institute of Solid-State Chemistry Ural Branch of Russian Academy of Science, 620049 Yekaterinburg, Russia
* Correspondence: shishkin@ihim.uran.ru; Tel.: +7-922-133-42-69

**Abstract:** The rock-salt ordered $A_2CuWO_6$ (A = Sr, Ba) with $I4/m$ space group and disordered $SrCu_{0.5}M_{0.5}O_{3-\delta}$ (M = Ta, Nb) with $Pm3m$ space group perovskites were successfully obtained via a solid-state reaction route. Heat treatment of $Ba_2CuWO_6$ over 900 °C in air leads to phase decomposition to the barium tungstate and copper oxide. Thermogravimetric measurements reveal the strong stoichiometric oxygen content and specific oxygen capacity ($\Delta W_o$) exceeding 2.5% for $Ba_2CuWO_6$. At the same time, oxygen content reveals $Cu^{3+}$ content in $SrCu_{0.5}Ta_{0.5}O_{3-\delta}$. Under the following reoxidation of $Ba_2CuWO_6$, step-like behavior in weight changes was observed, corresponding to possible $Cu^+$ ion formation at 900 °C; in contrast, no similar effect was detected for $M^{5+}$ cations. The yellow color of $Ba_2CuWO_6$ enables to measure the band gap 2.59 eV. $SrCu_{0.5}Ta_{0.5}O_{3-\delta}$ due to high oxygen valance concentration has a low thermal conductivity 1.28 $W·m^{-1}·K^{-1}$ in the temperature range 25–400 °C.

**Keywords:** perovskite oxide; high temperature redox; crystal structure; thermal properties; TGA

## 1. Introduction

Complex oxides with perovskite structure $ABO_3$ are well known because of the variety of properties enabling applications in different industrial and research areas. Such perovskites could be considered as semiconductors [1,2], dielectrics [3], thermoelectrics [4], thermal barrier coatings [5], optical and luminescence materials [6] and others. Perovskites with oxygen nonstoichometry and effective oxygen transport could be used as materials for heat storage or combustion. Due to dramatic climate changes, ecological friendly technologies are in great demand.

The green energy part in total heat power production is projected to increased annually until 2040 [7]. Nevertheless, in 2022 more than 60% of electricity was generated by fuel combustion: coal (34%), oil (38%) and natural gas (28%) [8]. Therefore, material design for ecologically friendly power generation technologies such as chemical looping with oxygen uncoupling (CLOU) is still currently required. The CLOU approach is based on the use of oxygen carriers (OCs) that uptake oxygen from the air for subsequent fuel combustion [9]. Then, the oxygen-depleted OC particles are transferred to the air reactor for reoxidation.

OC properties significantly affect the fuel conversion within the CLOU process; as a consequence, several requirements were considered [9–13]:

- The combination of temperature and equilibrium values of oxygen partial pressure should be sufficient to provide a complete conversion in a fuel reactor and fast oxygen saturation of OCs under oxidation.
- The reaction in the fuel reactor should be exothermic, providing some temperature increase, which promotes the release rate of gas-phase oxygen.
- Rather high oxygen capacity (>3%).
- Thermodynamic stability under reducing atmospheres specifically for the fuel reactor.
- Elevated kinetic parameters of oxygen exchange with ambient gas.





Cu-based OCs are of special interest due to the combination of high values of oxygen storage with fast oxygen exchange with the gas phase. However, reducing conditions are known to promote metallic copper formation, which tends to melt and agglomerate at elevated temperatures. Such an objectionable effect leads to a reduction in effective surface area of the particles used and as a consequence a decrease in combustion efficiency. Therefore, most research has paid attention to the design of composite materials where CuO is applied as the active component deposited on the inert material support such as alumina, zirconia, ceria [9,11,14,15].

Among the copper-based complex oxides, Ruddlesden–Popper compounds related to a $K_2NiF_4$-type structure are well recognized. These materials have demonstrated elevated catalytic activity, oxygen exchange rate and electrochemical performance as cathodes for solid oxide fuel cells with unusual magnetic properties [16]. Special attention is paid to the copper-based phase $La_2CuO_4$ [17]. These materials are also characterized by having enough stability with respect to crystal structure and functional characteristics at high temperatures but they can be completely decomposed in reducing media. On the other hand, complex oxides with perovskite structure are suggested as promising OCs because of their long-term mechanical stability, good fluidization properties, and appreciable reactivity with gaseous fuels [9,12,18–20]. It is interesting to combine the properties of copper-based oxides in a structured perovskite-like motif. For example, pure copper perovskite $LaCuO_{3-\delta}$ can be obtained only when an oxygen-rich atmosphere is applied; moreover, in CLOU processes it tends to decompose and mostly fails to reassemble within the reoxidation cycle [21,22]. Such an effect is connected with the presence of unstable $Cu^{3+}$ cations. The structure and charge stabilization could be achieved by a partial doping with high valence cation as $Ta^{5+}$, $Nb^{5+}$, $W^{6+}$. However, there is a lack of high temperature redox behavior and thermal stability of such perovskites.

Considering the aforementioned data, it is interesting to examine properties of copper-containing complex oxides and to conduct an initial assessment for its implementation as the OC material for the CLOU process [23]. In addition, some basic properties of the studied compounds such as thermal conductivity and expansion coefficient as well as a band gap value were estimated in order to extend existing characteristics. As the stable copper compound, the rock-salt ordered double $A_2CuWO_{6-\delta}$ where A = Ba or Sr and disordered $SrCu_{0.5}M_{0.5}O_{3-\delta}$ where M = Nb, Ta perovskite-type phases were chosen for the study.

## 2. Materials and Methods

### 2.1. Materials and Sample Preparation

The $A_2CuWO_{6-\delta}$ and $SrCu_{0.5}M_{0.5}O_{3-\delta}$ powders where A = Sr, Ba and M = Ta, Nb were synthesized via a conventional solid-state route. High-purity tungsten ($WO_3$ 99.999%), copper (CuO 99.99%), tantalum ($Ta_2O_5$, 99.999%), niobium ($Nb_2O_5$, 99.999%) oxides and barium ($BaCO_3$ 99.99%) and strontium ($SrCO_3$ 99.99%) carbonates were selected as raw materials. The starting reagents were weighed in required ratios and mixed for 30 min in ethanol media. After thorough grinding in agate mortar, the compositions were fired for 12 h at 870 °C in the air in an alumina crucible. The temperature was chosen because of the endothermic peak corresponding to the solid-state reaction observed at DSC earlier. The resulting powder was ground, pelletized with 50 MPa uniaxial pressure within 30 s at Hydraulic Press 660 (Silfradent, Italy) and annealed in a platinum crucible at 900–1150 °C for 12 h in the air. At each temperature, the sintered body was crushed and grinded in ethanol media in agate mortar, analyzed via XRD-technique, pelletized and annealed at a temperature 50 °C above the previous one.

### 2.2. Characterization

The phase composition and structural studies were conducted using a Shimadzu XRD 7000 (Shimadzu, Japan) diffractometer ($Cu_{K\alpha}$ radiation ($\lambda$ = 1.5418 Å) between angles from

10 to 80°, with a step of 0.03° and a scanning rate of 5 s per point. Diffraction patterns were collected from the polished surface of the sintered tablet form sample.

Thermogravimetric analysis of the sample (weighted with an accuracy of 0.01 mg) was carried out at Setaram TG-92 (Setaram, France) in air and argon flows. Isothermal dependence of mass change in Cu-based perovskites at variable atmosphere was recorded at consistent switching of gaseous fluxes as follows: air → (5% $H_2$ in Ar) → air. Switching of the gas medium was carried out via vacuuming the measuring cell and triple purging with gas in which measurements were to be taken. Oxygen content in the synthesized samples was determined via total reduction in the 5% $H_2$ in Ar mixture at 900 °C for $Ba_2CuWO_{6-\delta}$ or 950 °C for $SrCu_{0.5}Ta_{0.5}O_{3-\delta}$.

The scanning electron microscopy (SEM) images were obtained from JEOL JSM 6390LA (Jeol, Japan). Chemical composition of the obtained materials was controlled via the energy-dispersive X-ray spectroscopy (EDX) technique using a Jeol JED2300 EDX analyzer implemented to the microscope specified.

Perovskite structural predictors, such as tolerance (t), new tolerance (τ), octahedral (μ) factors and octahedral mismatch (Δμ) were calculated by approaches thoroughly described earlier [24–26]. Electronegativity and chemical hardness mismatch of B-cations were calculated using the data [27,28].

HSC Chemistry 7.0 was used to assess thermodynamical probability of the decomposition reactions.

Thermal conductivity was measured in the temperature range 25–150 °C at the custom-made installation IT-λ-400 discussed in detail earlier [29]. The experimentally obtained value was adjusted for the porosity value according to the known equation [30]:

$$\frac{\lambda_{exp}}{\lambda_{dense}} = 1 - \frac{4}{3} \cdot P \tag{1}$$

where $\lambda_{exp}$ and $\lambda_{dense}$ are experimental and porous free sample thermal conductivity, respectively, $W \cdot m^{-1} \cdot K^{-1}$, P is porosity of the specimen.

Thermal expansion measurements were carried out with a Linseis L75/1250 dilatometer (Linseis Messgerate, Germany) using rectangular cuts of dense samples with a heating/cooling rate of 5 K/min at 25–1000 °C.

Details for band gap measurements were thoroughly discussed in the paper [31].

### 3. Results and Discussion

*3.1. Synthesis and High-Temperature Redox Behavior of $Ba_2CuWO_{6-\delta}$*

It is known that copper forms rock-salt ordered double perovskites with a tetragonal structure (S.G. *I4/m*) with $W^{6+}$ as $A_2CuWO_6$, where A = Ba or Sr [32–37]. The double perovskites $A_2CuWO_6$ can be formed according to equation:

$$2ACO_3 + CuO + WO_3 = A_2CuWO_6 + 2CO_2^g \tag{2}$$

where A = Ba, Sr.

The formation mechanisms as well as high-temperature redox behavior are illustrated on the example of $Ba_2CuWO_6$. The results of XRD (Figure 1a) show that the mixture after the firing at 870 °C in air consists of mostly barium tungstates, $BaWO_4$, $Ba_2WO_5$, $Ba_3WO_6$, $Ba_3W_2O_9$ and copper-containing oxides $Ba_2CuWO_6$, CuO and $Ba_2Cu_2O_5$. Despite the observed solid-state reaction initiation temperature at DSC analysis, it was suggested to slightly increase the synthesis temperature to accelerate the reaction rate. Hence, the following manual grinding in ethanol media and pressing with heat treatment at 900 °C contributes to active interplay of the components enabling the desired oxide to be obtained after holding at the synthesis temperature for 50 h. The XRD pattern of the as-prepared sample is shown in Figure 1b. All the reflections observed can be reliably ascribed to tetragonal $Ba_2CuWO_6$ having the *I4/m* space group, indicating the single-phase formation. The

refined values for unit cell parameters in $Ba_2CuWO_6$ and $Sr_2CuWO_6$ as well as perovskite structure predictors are collected in Table 1.

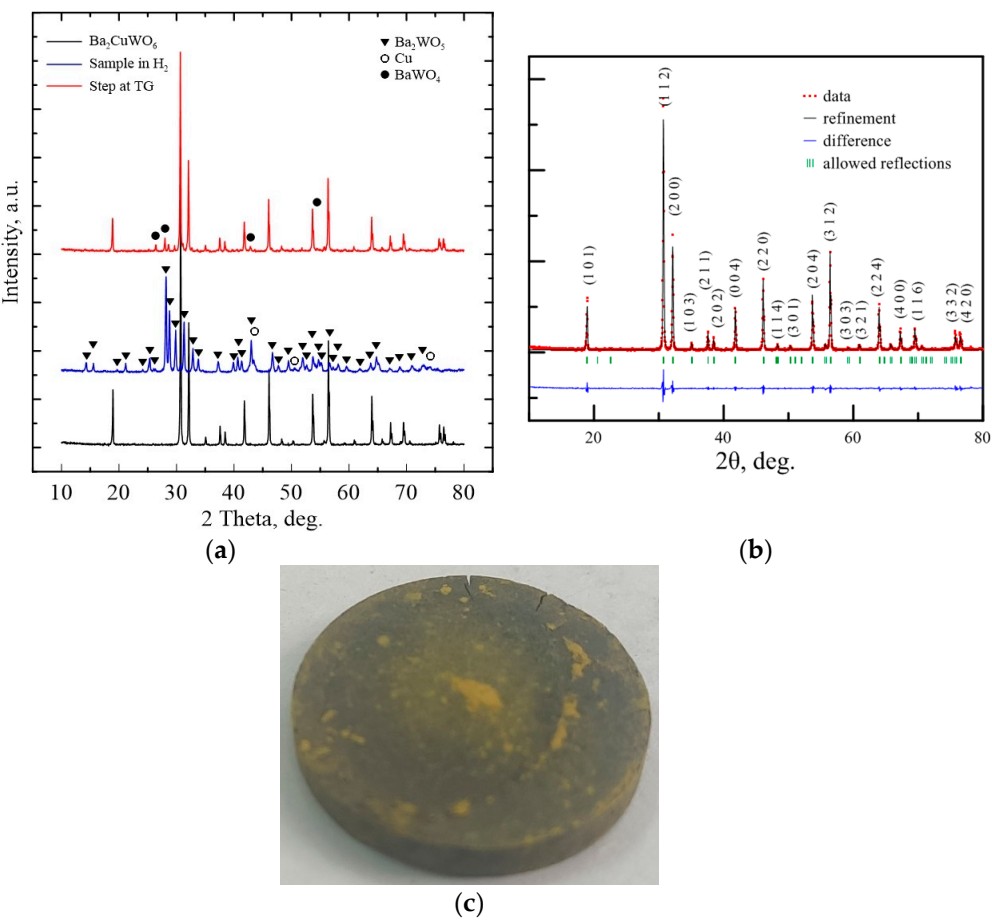

**Figure 1.** The XRD patterns of (**a**) samples synthesized at different temperatures; (**b**) single-phase $Ba_2CuWO_{6-\delta}$ obtained after 900 °C; (**c**) image of the blackened sample after 950 °C.

**Table 1.** Structural and thermodynamical parameters of Cu-based perovskites.

| | $Ba_2CuWO_6$ | $Sr_2CuWO_6$ | $SrCu_{0.5}Ta_{0.5}O_{3-\delta}$ | $SrCu_{0.5}Nb_{0.5}O_{3-\delta}$ |
|---|---|---|---|---|
| $a \pm 0.03$, Å | 5.562 | 5.433 | 3.976 | 3.978 |
| $c \pm 0.03$, Å | 8.630 | 8.382 | - | - |
| t | 1.039 | 0.979 | 0.969 | 0.969 |
| μ | 0.493 | 0.493 | 0.507 | 0.507 |
| Δμ | 0.048 | 0.048 | 0.033 | 0.033 |
| τ | 3.636 | 3.506 | 3.630 | 3.630 |
| Δχ (Pauling) | 0.46 | 0.46 | 0.4 | 0.3 |
| Δχ (Mulliken) | 0.3 | 0.3 | 0.2 | 0.5 |
| Δχ (Allen) | 0.38 | 0.38 | 0.51 | 0.44 |
| Δ (Chemical hardness) | 0.2 | 0.2 | 0.7 | 0.5 |

Despite the literature data [34–36], it was found (Figure 1a) that calcination in the range 950–1200 °C leads to the $Ba_2CuWO_6$ phase partial decomposition: the black shade at the yellow-colored sample is clearly observed, which may result from the formation of copper (II) oxide on the surface. The image of blackened sample after 950 °C annealing is

presented in Figure 1c. Interestingly, the sample cross-section is also black, which suggests that neither air nor crucible material affects the decomposition.

$Ba_2WO_5$ presence could be explained by the equation of $Ba_2CuWO_6$ decomposition:

$$Ba_2CuWO_6 = Ba_2WO_5 + CuO \qquad (3)$$

Two phases $Ba_2CuWO_6$ and $Ba_2WO_5$ were observed to provide after the calcination at 950 °C; barium tungstate forms according to decomposition reaction (3) and is indirectly proved by the image (Figure 1c). The formed copper oxide was not detected via the XRD technique, but a combination of SEM with EDX data partially confirms the mechanism suggested (Figure 2). As can be seen from Figure 2b, dark color grains presumably correspond to copper oxide, while light grains apparently are barium tungstate. The decomposition process (Equation (3)) takes place in the whole volume (Figure 2a) of the cross-section view of the sample. Such an effect reveals pretty low thermal stability of $A_2CuWO_6$ phases within the synthesis process applied.

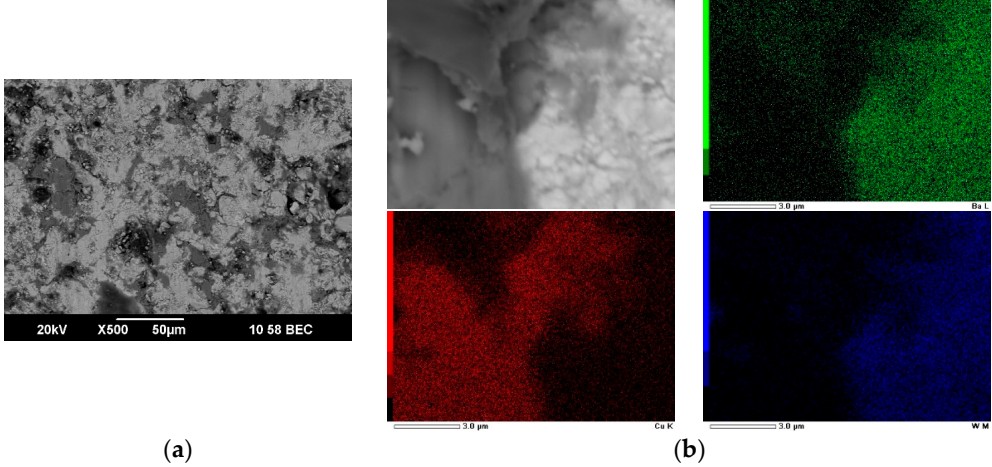

(**a**)                (**b**)

**Figure 2.** (**a**) BEC image of cross-section of $Ba_2CuWO_{6-\delta}$ synthesized at 950 °C (**b**) mapping of decomposition border: Ba—green; Cu—red; W—blue.

Moreover, heating of the single-phase $Ba_2CuWO_6$ up to 1000 °C results in $BaWO_4$ impurity formation. Interestingly, no considerable heat effect of the decomposition process at DSC equipment (Setsys Evo-18, Setaram, France) was observed. The formation mechanism of $BaWO_4$ within heating over 900 °C could be represented by Equation (4). To estimate barium carbonate presence, EDX analysis was applied. Unfortunately, the EDX technique fails to assess light element quantity, such as carbon, but it could help to find barium-containing phase without the presence of copper or tungsten. The upper right corner of the EDX mapping (Figure 2b) shows high barium content, while the copper and tungsten amount are low. That area is supposed to be $BaCO_3$, resulting from $Ba_2WO_5$ decomposition. Barium oxide formation within any decomposition reaction is not thermodynamically favorable in air. According to the thermodynamical data, reaction (4) could take place only on cooling of the sample lower than 442 °C. The hypothesis was proved via rapid cooling of the sample in the liquid nitrogen: no peaks at XRD pattern of the sample corresponding to the $BaWO_4$ were observed.

The following temperature enhancement yields $Ba_2WO_5$ and $BaWO_4$ impurity accumulation from 2.3% at 950 °C to 6.9% at 1250 °C. The observed phenomenon evidences higher thermodynamical stability of barium tungstates than copper-based double perovskite. This could be connected with a high value of tolerance factor (1.039, Table 1) for $Ba_2CuWO_6$, which reveals high stretching of the lattice [24].

$$Ba_2WO_5 + CO_2^g = BaWO_4 + BaCO_3 \quad \Delta G^0 = -11.315 \text{ kJ} \quad \text{at } 400\,°C \qquad (4)$$

Both oxygen content and high-temperature redox behavior were examined via TGA under gas media containing 5% $H_2$ in Ar (Figure 3a). Reducing atmosphere at 900 °C $Ba_2CuWO_6$ easily decomposes through the following equation:

$$Ba_2CuWO_6 + H_2^g = Ba_2WO_5 + Cu + H_2O^g \tag{5}$$

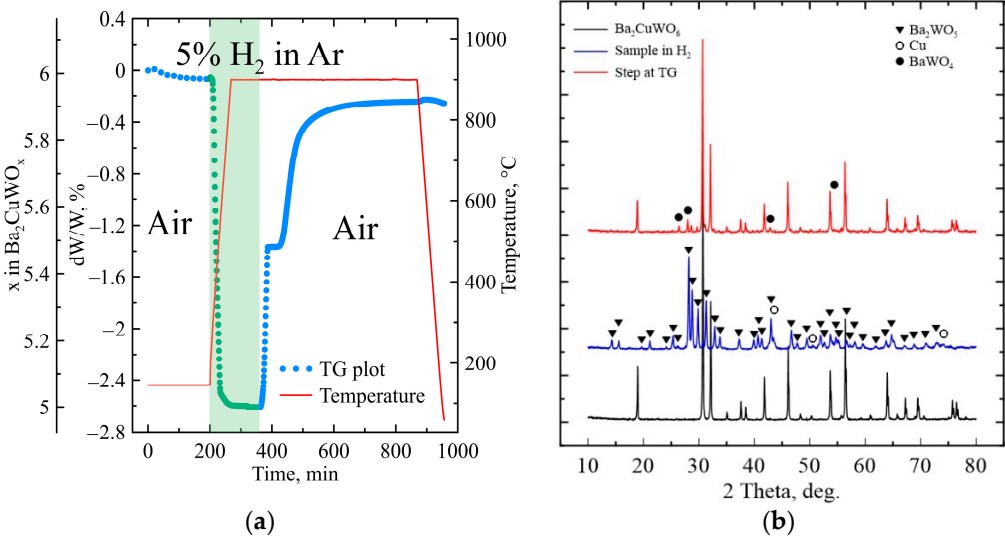

**Figure 3.** (**a**) Isothermal dependence of mass change in $Ba_2CuWO_{6-\delta}$ at the variable atmosphere; (**b**) XRD patterns of the sample after TGA.

The TGA results show (Figure 3a) that in contrast to the other copper-containing mixed oxides [12,23], $Ba_2CuWO_6$ reoxidizes with an immediate step corresponding to the $Ba_2CuWO_{5.55}$, where copper cations nominally accord with Cu(I) state. A single-charged copper has a slightly bigger ionic radii compared with $Cu^{2+}$ 0.77 and 0.73 Å correspondingly [38]. That leads to reducing the tolerance factor value to 1.029 and a partial lattice distortion decrease. Moreover, due to high charge of the tungsten (+6), the overall B-site charge remains at the sufficient level (+3.5) to maintain the perovskite-like structure. Together, both the above-mentioned factors result in appearance of the step during the reoxidation process.

The same nonstoichiometry ($Ba_2CuWO_{5.58}$) could be achieved in an argon atmosphere: the value of media oxygen partial pressure allowing one to uncouple the oxygen and reduce the copper from +2 to +1 state. After one reoxidation cycle, $Ba_2CuWO_6$ fails to obtain a single-phase sample (Figure 3b).

### 3.2. Synthesis and High-Temperature Redox Behavior of $SrCu_{0.5}Ta_{0.5}O_{3-\delta}$

The effect of fivefold charged cations such as $Nb^{5+}$, $Ta^{5+}$ on the copper-based perovskite-like structures formation and their properties was studied. The obtained compounds obey disordered perovskite structure contrary to rock-salt ordered double perovskite observed for tungsten-based oxide. However, the cuprates containing $M^{5+}$ are characterized by higher symmetry, since both $SrCu_{0.5}Ta_{0.5}O_3$ and $SrCu_{0.5}Nb_{0.5}O_3$ are found to crystallize with a cubic *Pm3m* crystal structure. An example of such a Rietveld refined XRD pattern is illustrated in Figure 4a. The grains with a cubic symmetry could also be clearly observed at the SEM images (Figure 5).

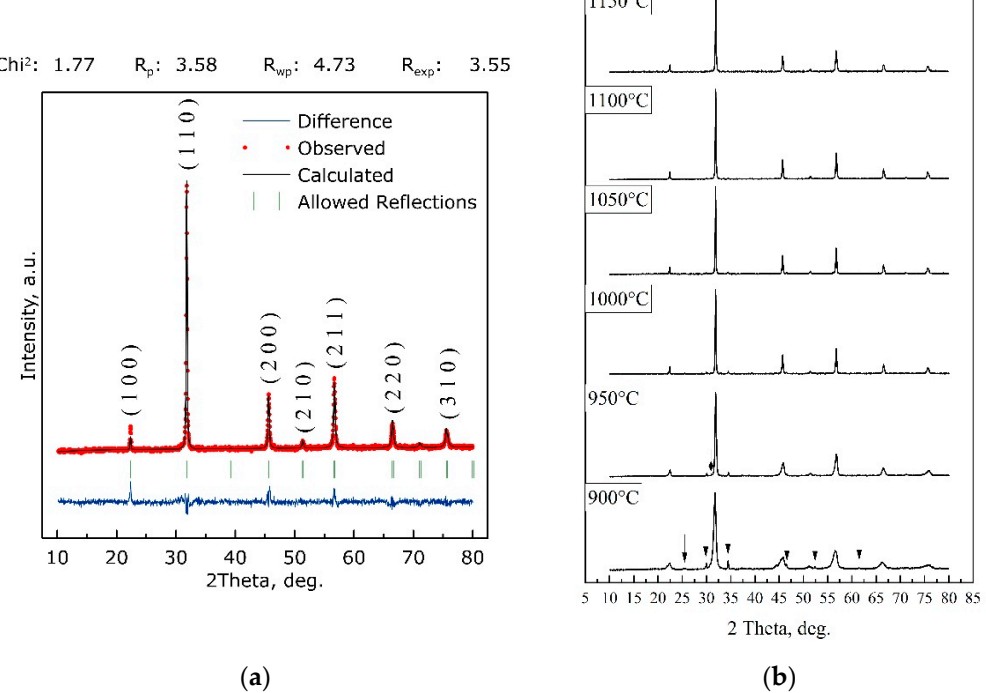

(**a**)  (**b**)

**Figure 4.** The XRD pattern of (**a**) $SrCu_{0.5}Ta_{0.5}O_{3-\delta}$ with the results of Rietveld refinement; (**b**) samples synthesized at different temperatures: arrow—$SrO \cdot CuO$; diamond—$Sr_2Ta_2O_7$.

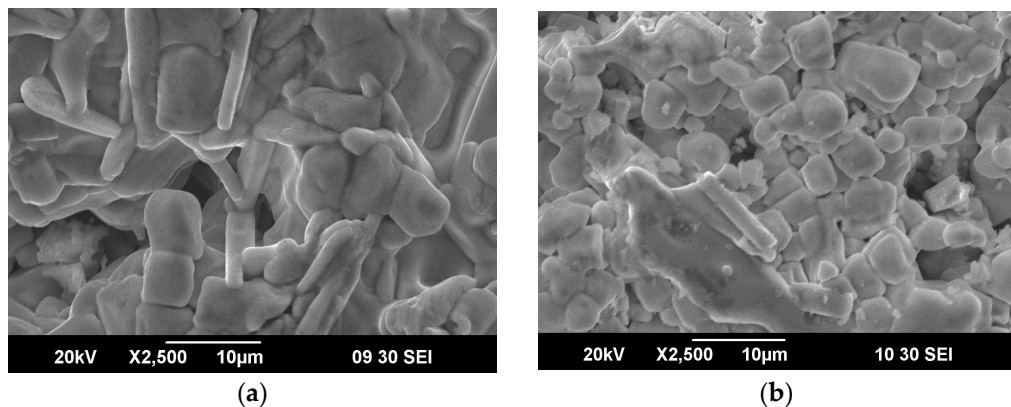

(**a**)  (**b**)

**Figure 5.** The SEM images of $SrCu_{0.5}M_{0.5}O_{3-\delta}$ for M = (**a**) Ta, (**b**) Nb.

Moreover, Ta and Nb containing perovskites possess higher thermal stability. The cubic phase appears at 900 °C (Figure 4b) with a mixed oxide $CuO \cdot SrO$ and $Sr_2M_2O_7$ as impurities. The formation mechanism could be expressed by the following reactions:

$$SrCO_3 + CuO = SrO \cdot CuO + CO_2^g \tag{6}$$

$$2SrCO_3 + Nb_2O_5 = Sr_2Nb_2O_7 + 2CO_2^g \tag{7}$$

$$Sr_2Nb_2O_7 + 2SrO \cdot CuO = 4SrCu_{0.5}Nb_{0.5}O_{2.75} \tag{8}$$

The proposed mechanism is partially confirmed via EDX (Figure 6) analysis applied for the sample preheated at 1000 °C. The final perovskite phase grains are separated via melted phase $CuO \cdot SrO + CuO$ ($T_{m.p.}$ (CuO) = 975 °C, $T_{m.p.}$ ($SrCuO_2$) = 1080 °C [39], points 006–010 in Figure 6), which in contact with $Sr_2M_{2-x}O_7$ (M = Ta, Nb) forms the desirable

compound. The samples containing only Cu-based perovskite phase $SrCu_{0.5}M_{0.5}O_3$ were obtained after 1150 °C and their crystal cell parameters were refined via the Rietveld method (Table 1).

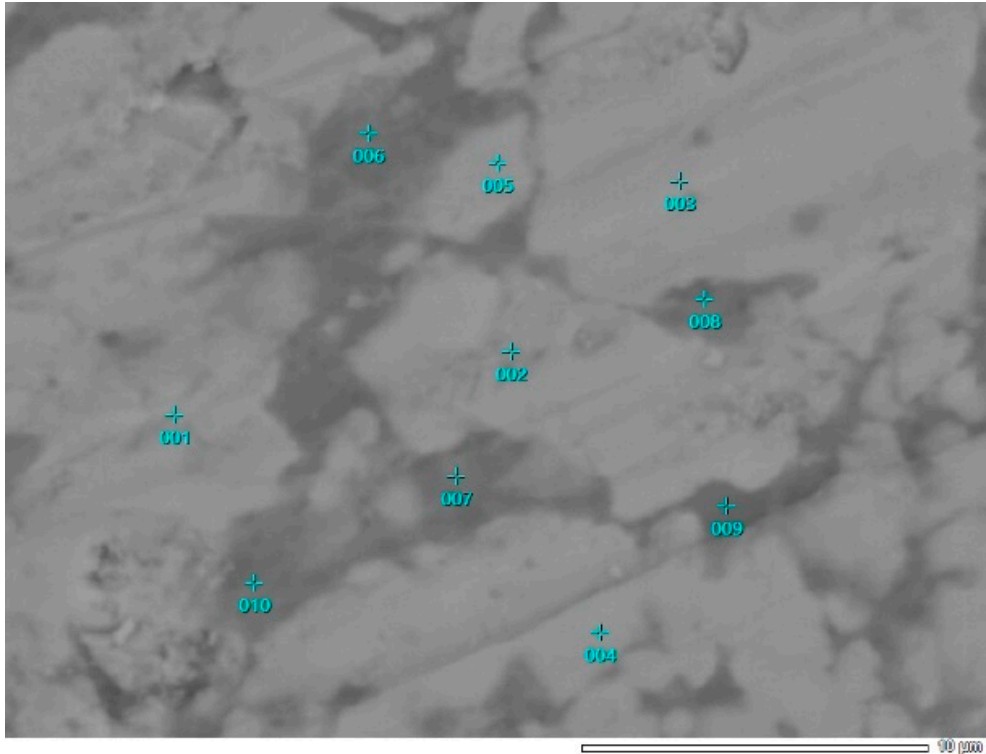

**Figure 6.** EDX point analysis of Sr-Cu-Ta-O system calcinated at 1000 °C for 24 h.

According to the EDX results (Table 2), there is a slight deviation from the stoichiometric ratio of copper and tantalum (3:2), which was not observed in [40]. At the same time, the presence of tantalum in the binding phase of SrO·CuO can be explained by the formation of $Sr_2Ta_2O_7$, the amount of which is insufficient for its identification via the XRD method. It has been shown [40] that under equilibrium conditions at 950 °C of the $SrO-CuO-TaO_{2.5}$ system with an atomic ratio of 2:1:1, two phases are present: $SrCuO_2$ and $Sr_3Ta_{2-x}Cu_{1+x}O_{9+\delta}$. This is confirmed by the proposed mechanism (7): there is a gradual dissolution of complex copper and strontium oxide in strontium tantalate. However, in contrast to the presented work, the formation of cubic perovskite is observed for both Ta and Nb, including at 950 °C, which was also presented in a number of papers [41–43]. A significant deviation in the content of elements (Sr, Cu, Ta) at points 006–010 is caused by the migration of cations in the SrO·CuO melt at the synthesis temperature. $Sr_2Ta_2O_7$ particles in the crystallized melt were not detected via the SEM method. Their presence is only indirectly confirmed via EDX and XRD data.

Redox behavior of $SrCu_{0.5}Ta_{0.5}O_{3-\delta}$ (Figure 7) also completely differs from $Ba_2CuWO_6$. After applying a single red-ox cycle, no weight changes compared with initial state were clearly seen. No intermediate steps within sample reoxidation were observed. Moreover, there is considerable oxygen nonstoichometry, which corresponds to $SrCu_{0.5}Ta_{0.5}O_{2.81}$ wherein the oxygen loss rate is twice as low (time for almost complete reduction) but twice as high as the reoxidation rate. Finally, oxygen content 2.75 corresponds to the presence of copper in $Cu^{2+}$ state, while 2.81 suggests the appearance of $Cu^{3+}$.

**Table 2.** Sr-Cu-Ta-O system calcinated at 1000 °C EDX analysis results.

| N | Sr, at. % | Cu, at. % | Ta, at. % |
|---|---|---|---|
| 001 | 47.66 | 30.73 | 21.61 |
| 002 | 47.03 | 29.58 | 23.39 |
| 003 | 45.76 | 32.03 | 22.21 |
| 004 | 46.1 | 32.31 | 21.59 |
| 005 | 48.7 | 28.66 | 22.64 |
| 006 | 52.66 | 40.59 | 6.75 |
| 007 | 64.1 | 25.37 | 10.53 |
| 008 | 45.71 | 41.01 | 13.27 |
| 009 | 55.8 | 29.27 | 14.94 |
| 010 | 45.47 | 44.54 | 9.89 |

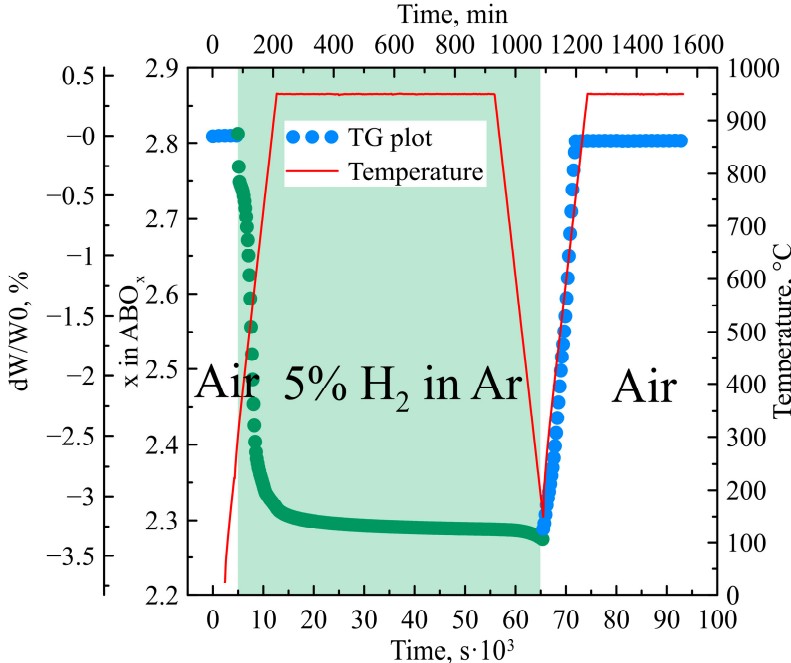

**Figure 7.** Isothermal dependence of mass change in $SrCu_{0.5}Ta_{0.5}O_{3-\delta}$ at the variable atmosphere.

The lowest thermal stability of $Ba_2CuWO_6$ among considered compounds could be connected with a high tolerance factor (t, Table 1) close to the perovskite stability edge. Even large Cu+ cation (0.77Å) in the intermediate state $Ba_2CuWO_{5.545}$ does not permit considerable reduction in lattice distortions (t = 1.029). The other structural parameters (Table 1) are close for all synthesized Cu-based perovskites and lay into the region of stable perovskite structure. The electronegativity mismatch in all considered scales shows similar results for $W^{6+}$ and $Ta^{5+}$ or $Nb^{5+}$, revealing alike kinds of bonds in nature. Interestingly, a low chemical hardness mismatch between copper and tungsten shows closer affinity to electrons of Cu and W than with Nb or Ta.

### 3.3. Some Properties of Cu-Based Perovskites

Due to the low thermal stability, which makes it complicated to produce well sintered ceramics, tungsten-based cuprates could be considered as luminescence materials and coatings [44]. The yellow color of $Ba_2CuWO_6$ allows one to suggest a considerable band gap value which amounted to 2.59 eV (Figure 8a), so this material should be characterized

as a wide gap semiconductor with promising base for the photocatalytic water splitting approach and can be suggested as an alternative for titania in this field.

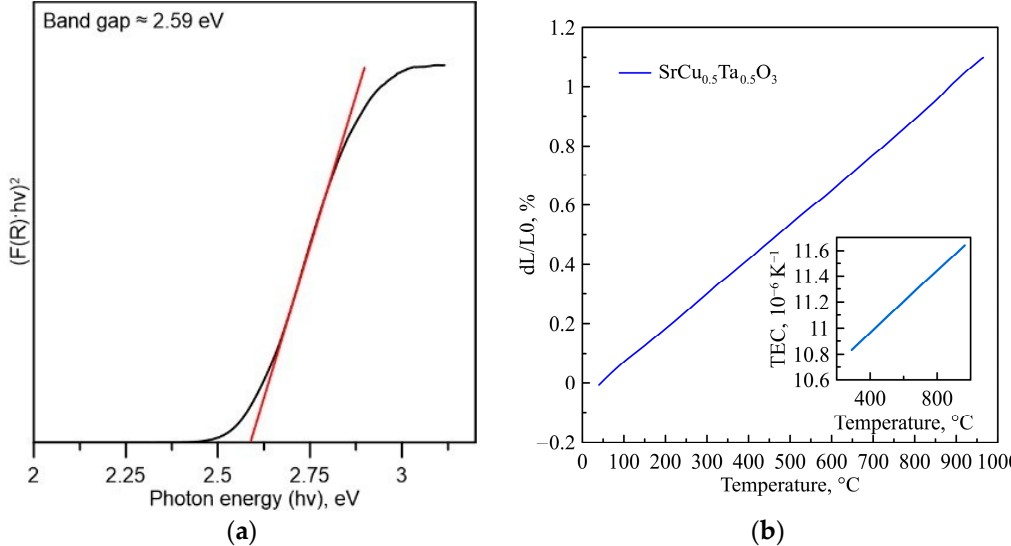

**Figure 8.** Plot of (**a**) (R·hυ)2 as a function of photon energy for the estimation of direct energy band gap; (**b**) thermal expansion of $SrCu_{0.5}Ta_{0.5}O_{3-\delta}$.

A rather high thermal stability of $SrCu_{0.5}Ta_{0.5}O_{3-\delta}$ for Cu-based perovskite enables one to investigate high temperature properties such as thermal expansion. It was found that TEC (Figure 8b) gradually increases with a temperature growth and its value has average values of $10.8–11.6·10^{-6}\ K^{-1}$. The values obtained are believed to be appropriate for the use of tantalum-doped oxide in contact with other oxide materials at elevated temperatures. In addition, low thermal conductivity $1.28\ W·m^{-1}·K^{-1}$, attributed to high oxygen vacancy concentration, at temperatures 25–400 °C looks promising for thermoelectrical materials. Considerable oxygen exchange parameters and good thermal stability could assess the material for oxygen adsorption from gaseous media.

Finally, it should be stressed here that a light doping of copper-based perovskites with highly charged compounds does not sufficiently increase phase stability. As can be seen even a simple heating in air is accompanied mostly by a partial decomposition with copper oxide crystallization. This fact constrains the application of the studied materials in high-temperature devices, but taking into account elevated rate of oxygen exchange these materials could be proposed as oxygen carrier materials and oxygen sorbents at lower temperatures.

### 4. Conclusions

- The rock-salt ordered double perovskites $A_2CuWO_{6-\delta}$ (A = Sr, Ba) with the *I4/m* space group and disordered perovskites $SrCu_{0.5}M_{0.5}O_{3-\delta}$ (M = Nb, Ta) with a *Pm3m* space group were synthesized via a solid-state reaction route.
- The structural predictors for the synthesized Cu-based perovskites were calculated and the cell parameters were refined via the Rietveld method.
- Redox behavior of $Ba_2CuWO_6$ was studied at 900 °C and a step was found within reoxidation which should contribute to the presence of Cu(I).
- $Ba_2CuWO_6$ decomposes after 900 °C with the formation of copper oxide and barium tungstanate; the suggested mechanism was approved via EDX analysis.
- The value of the measured $Ba_2CuWO_6$ band gap was 2.59 eV.
- The disordered perovskite $SrCu_{0.5}M_{0.5}O_{3-\delta}$ (M = Nb, Ta) forms within the reaction of the liquid complex strontium-copper oxide and strontium niobate (tantalate) at temperatures in the range 900–1150 °C.

- The copper-based perovskite compounds with $M^{5+}$ have to consist of $Cu^{3+}$ according to the oxygen content.
- Thermal properties of $SrCu_{0.5}Ta_{0.5}O_{3-\delta}$ were investigated: average TEC value at 1000 °C $11.6 \cdot 10^{-6}$ $K^{-1}$ and a low thermal conductivity 1.28 $W \cdot m^{-1} \cdot K^{-1}$ in the temperature range 25–400 °C.

**Author Contributions:** Conceptualization, R.A.S. and A.Y.S.; methodology, R.A.S.; validation, R.A.S.; formal analysis, R.A.S.; investigation, R.A.S. and M.O.K.; resources, R.A.S. and A.Y.S.; data curation, R.A.S. and A.Y.S.; writing—original draft preparation, R.A.S.; writing—review and editing, R.A.S. and A.Y.S.; visualization, R.A.S.; supervision, R.A.S.; project administration, R.A.S. All authors have read and agreed to the published version of the manuscript.

**Funding:** The author appreciates the support of this work by the Russian Science Foundation under grant 22-19-00129.

**Institutional Review Board Statement:** Not applicable.

**Informed Consent Statement:** Not applicable.

**Data Availability Statement:** Not applicable.

**Conflicts of Interest:** The authors declare no conflict of interest.

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
