# Peer review of "Phase Instability, Oxygen Desorption and Related Properties in Cu-Based Perovskites Modified by Highly Charged Cations"

_ceramics, doi:10.3390/ceramics6020057_

Round 1

Reviewer 1 Report

Line 80 "were fired for 12 hours 80 at 870 °C"

What justifies the choice of this temperature for precalcination?

Table 1
How the such accurate values of the structural parameters were obtained?

How was the oxidation state of copper and other elements determined in the samples (or only oxygen non-stoichiometry calculations were used)?

Author Response

Dear reviewers, thank you for your work and appreciation of the manuscript assessment. All suggested comments were considered during the major revision of the paper.

Reviewer 1

Line 80 "were fired for 12 hours 80 at 870 °C"

What justifies the choice of this temperature for precalcination?

The temperature was chosen because of endothermic peak corresponding to the solid-state reaction observed at DSC earlier.

Table 1
How the such accurate values of the structural parameters were obtained?

The cell parameters were refined by the Rietveld Refinement, the error values were added.

How was the oxidation state of copper and other elements determined in the samples (or only oxygen non-stoichiometry calculations were used)?

Copper oxidation state was calculated from the oxygen content determined by TGA analysis.

Reviewer 2 Report

The article presents an exciting research investigation into how highly charged cations impact the structure and properties of Cu-based perovskites. The researchers utilized a solid-state reaction method to obtain two different types of perovskites - rock-salt ordered A2CuWO6 and disordered SrCu0.5M0.5O3-δ and then analyzed their thermal properties, stoichiometric oxygen content, specific oxygen capacity, band gap, and thermal conductivity. Additionally, the study explored the redox behavior of Ba2CuWO6 and identified the presence of Cu(I).

To further improve the manuscript, the authors are recommended to address the following comments and incorporate them accordingly.

1)      To enhance the article's clarity and coherence, it would be beneficial to use simpler language and vary sentence lengths. Furthermore, clearer transitions between ideas would improve the manuscript's readability and comprehension. A few suggestions are given below:

i)                    Consider changing "strongly stoichiometric" to simply "stoichiometric" or “strong stoichiometric”

ii)                   In line 22, consider using "variety of properties" instead of "high verielty of properties" or simply "various properties."

iii)                 In line 27-28, consider revising to “Due to dramatically climate changes ecological friendly technologies are in great demand.” It should be either “due to dramatic or drastic change” or “ The increasing impact of climate change has led to a high demand for eco-friendly technologies”.

iv)                 Line 81# “Consider changing “ground,” to “grinded”

v)                   Line 127# dot is missing.

vi)                 Line 125# consider revising the sentence to provide more context and clarify the meaning of "an additional seria of cuprates”.

vii)               Line 207# consider revising the sentence to clarify the meaning: “The single-phase samples were obtained after 1150°C and crystal structure calcination”

and several other sentences also need to rewrite throughout the manuscript.

2)      Authors are advised to elaborate on the reasons for observing step-type behavior in the TGA curve when replacing Air with Ar+5%H2?

3)      Line 25# Add reference for "Such perovskites could be considered as semiconductors……"

4)      Line 30# Provide a reference for the statement "The green energy part in total heat power production is estimated to be increased annually until 2040".

5)      Line 30-31# Authors are recommended to provide some recent data after 2020 in place of 2018, as the growth in green energy has led to a decrease in fuel combustion for generating electrical energy. This would update the manuscript with the latest information and further support the points made in the text.

6)      Line 82# Authors are advised to explain “intermediate intensive grindings” Means how the intermediate grinding was performed? For example, was the sample heated in a furnace, allowed to cool down, then ground and placed back for heat treatment, or during heating at 900-1150, how intermediate grinding has been carried out? Additionally, it would be helpful to know if the grinding was done manually or using an instrument, what parameters were used, and how the grinding affected the microstructures of the final sample.

7)      Line 128#  Clarify about “thorough grinding”. For instance, the grinding is done manually for XYZ time or using a machine.

8)      line 128#, it would be helpful to provide information on the parameters used for pellet formation this could help in the reproducibility of the study. This could include details such as the type of equipment used, the pressure applied during formation, or any other relevant factors that may have influenced the pellet formation process.

9)      Line 129# Please provide clarification regarding the “50-hour exposure” Specifically, was it a 50-hour heating period, a 50-hour exposure to air, or something else?

10)   Line 137# Authors are recommended if possible to provide a picture showing a black shade at the yellow-colored sample for the Cu(II) Oxide on the surface.

11)   Kindly explain how grain formation occurs only from BaCO3 as mentioned in EDX data shown in fig 2? It would be helpful to explain this more thoroughly from both a qualitative and quantitative perspective.

12)   lines 177-178#, Kindly explain how the author calculated the oxygen difference up to the third decimal point. Additionally, it would be helpful to know how significantly this third decimal point of the oxygen content affects the desired perovskite properties.

13)   In Figure 3, could you please clarify what the red line represents?  Additionally, it would be helpful to know if there was a sudden change from air to Ar+5%H2 gas purging into the furnace.

14)   In fig 6, the Authors are advised to provide one table mentioning the compositional percent analysis for all 10 points of EDX data corresponding to all elements which could be useful to identify the final compound composition for both phases.

15)   Regarding Figure 7, please provide information on the medium used during the initial 5 minutes which is not mentioned in the figure.

16)   Reference formatting needs to correct: ref 6,9, 13, 15,16, 19,20, 21,24, 25, 28,

Author Response

Dear reviewers, thank you for your work and appreciation of the manuscript assessment. All suggested comments were considered during the major revision of the paper.

Reviewer 2

The article presents an exciting research investigation into how highly charged cations impact the structure and properties of Cu-based perovskites. The researchers utilized a solid-state reaction method to obtain two different types of perovskites - rock-salt ordered A2CuWO6 and disordered SrCu0.5M0.5O3-δ and then analyzed their thermal properties, stoichiometric oxygen content, specific oxygen capacity, band gap, and thermal conductivity. Additionally, the study explored the redox behavior of Ba2CuWO6 and identified the presence of Cu(I).

To further improve the manuscript, the authors are recommended to address the following comments and incorporate them accordingly.

1)      To enhance the article's clarity and coherence, it would be beneficial to use simpler language and vary sentence lengths. Furthermore, clearer transitions between ideas would improve the manuscript's readability and comprehension. A few suggestions are given below:

  1. i)                    Consider changing "strongly stoichiometric" to simply "stoichiometric" or “strong stoichiometric”

A corresponding correction has been made

  1. ii)                   In line 22, consider using "variety of properties" instead of "high verielty of properties" or simply "various properties."

A corresponding correction has been made

iii)                 In line 27-28, consider revising to “Due to dramatically climate changes ecological friendly technologies are in great demand.” It should be either “due to dramatic or drastic change” or “ The increasing impact of climate change has led to a high demand for eco-friendly technologies”.

A corresponding correction has been made

  1. iv)                 Line 81# “Consider changing “ground,” to “grinded”

A corresponding correction has been made

  1. v)                   Line 127# dot is missing.

A corresponding correction has been made (line 129 dot was missing).

  1. vi)                 Line 125# consider revising the sentence to provide more context and clarify the meaning of "an additional seria of cuprates”.

The sentence was clarified: “The effect of fivefold charged cations such as Nb5+, Ta5+ on the copper-based perovskite-like structures formation and their properties was studied.”

vii)               Line 207# consider revising the sentence to clarify the meaning: “The single-phase samples were obtained after 1150°C and crystal structure calcination”

and several other sentences also need to rewrite throughout the manuscript.

The sentence was corrected: “The samples containing only Cu-base perovskite phase SrCu0.5M0.5O3 were obtained after 1150°C and their crystal cell parameters were refined by Rietveld method (table 1).”

2)      Authors are advised to elaborate on the reasons for observing step-type behavior in the TGA curve when replacing Air with Ar+5%H2?

The step-like behavior connected with a lattice distortions reduction within one valence copper presence. A single-charged copper has a slightly bigger ionic radii compared with Cu2+ 0.77 and 0.73 Å correspondingly[38]. That leads to reducing the tolerance factor value to 1.029 and a partial lattice distortion decrease. Moreover, due to high charge of the tungsten (+6), the overall B-site charge remains at the sufficient level (+3.5) to maintain the perovskite-like structure. Together both mentioned above factors results in appearance of the step during reoxidation process.

3)      Line 25# Add reference for "Such perovskites could be considered as semiconductors……"

The references were added.

4)      Line 30# Provide a reference for the statement "The green energy part in total heat power production is estimated to be increased annually until 2040".

The corresponding reference was added.

5)      Line 30-31# Authors are recommended to provide some recent data after 2020 in place of 2018, as the growth in green energy has led to a decrease in fuel combustion for generating electrical energy. This would update the manuscript with the latest information and further support the points made in the text.

The recent data with a reference was added.

6)      Line 82# Authors are advised to explain “intermediate intensive grindings” Means how the intermediate grinding was performed? For example, was the sample heated in a furnace, allowed to cool down, then ground and placed back for heat treatment, or during heating at 900-1150, how intermediate grinding has been carried out? Additionally, it would be helpful to know if the grinding was done manually or using an instrument, what parameters were used, and how the grinding affected the microstructures of the final sample.

After each temperature the sintered body was crushed and grinded in ethanol media in agate mortar, analyzed by XRD-technique, pelletized and annealed at a temperature 50°Ð¡ above the previous one.

7)      Line 128#  Clarify about “thorough grinding”. For instance, the grinding is done manually for XYZ time or using a machine.

The grinding was carried out manually in agate mortar with a pestle in an ethanol media within 30 minutes.

8)      line 128#, it would be helpful to provide information on the parameters used for pellet formation this could help in the reproducibility of the study. This could include details such as the type of equipment used, the pressure applied during formation, or any other relevant factors that may have influenced the pellet formation process.

The experimental description was updated: “The resulting powder was grinded, pelletized with 50 MPa uniaxial pressure within 30 sec at Hydraulic Press 660 (Silfradent, Italy), and annealed in a platinum crucible at 900-1150°C for 12 hours in the air.”

9)      Line 129# Please provide clarification regarding the “50-hour exposure” Specifically, was it a 50-hour heating period, a 50-hour exposure to air, or something else?

The “exposure” was replaced by “holding at the synthesis temperature for 50 h”

10)   Line 137# Authors are recommended if possible to provide a picture showing a black shade at the yellow-colored sample for the Cu(II) Oxide on the surface.

The image of blacked sample was added (fig. 1c).

11)   Kindly explain how grain formation occurs only from BaCO3 as mentioned in EDX data shown in fig 2? It would be helpful to explain this more thoroughly from both a qualitative and quantitative perspective.

To estimate barium carbonate presence EDX analysis was applied. Unfortunately, the EDX technique fails to assess light elements quantity, such as carbon, but it could help to find barium-containing phase without presence of copper or tungsten. Upper right corner of the EDX mapping (fig. 2b) shows high barium content, while copper and tungsten amount are low. That area is supposedly to be BaCO3, resulted from Ba2WO5 decomposition. Barium oxide formation within any decomposition reaction is not thermodynamically favorable in air. According to the thermodynamical data the reaction (4) could take place only within cooling of the sample lower 442°C. The hypothesis was proved by rapid cooling of the sample in the liquid nitrogen: no peaks at XRD pattern of the sample corresponding to the BaWO4 was observed.

12)   lines 177-178#, Kindly explain how the author calculated the oxygen difference up to the third decimal point. Additionally, it would be helpful to know how significantly this third decimal point of the oxygen content affects the desired perovskite properties.

The oxygen content was rounded to the second decimal point.

13)   In Figure 3, could you please clarify what the red line represents?  Additionally, it would be helpful to know if there was a sudden change from air to Ar+5%H2 gas purging into the furnace.

The legend was added to the TGA-plots. Switching of the gas medium was carried out by vacuuming the measuring cell and triple purging with gas in which measurements will be made.

14)   In fig 6, the Authors are advised to provide one table mentioning the compositional percent analysis for all 10 points of EDX data corresponding to all elements which could be useful to identify the final compound composition for both phases.

The corresponding table was added.

15)   Regarding Figure 7, please provide information on the medium used during the initial 5 minutes which is not mentioned in the figure.

The Air medium was added at the figure 7.

16)   Reference formatting needs to correct: ref 6,9, 13, 15,16, 19,20, 21,24, 25, 28,

The reference list was revised.

Round 2

Reviewer 2 Report

In the updated manuscript, the authors have made noteworthy enhancements. The writing is now clear and well-articulated, and the authors have considered all of my comments in a comprehensive manner. It is my conviction that the revised version of the manuscript will be highly appealing to the readership of the journal.